# Equivalences between network modularity and diverse low-dimensional representations

**Mika Rubinov**
Department of Biomedical Engineering
Vanderbilt University
Nashville, TN 37235
`mika.rubinov@vanderbilt.edu`

## Abstract

Modularity is a popular clustering objective in network science. Here, we equate normalized and generalized versions of the modularity with variants of $k$-means objective, spectral manifold learning objectives, and UMAP. These equivalences naturally lead to definitions of new representation objectives. As an example, we show that one of these objectives embeds brain-imaging data much better than UMAP. Together, our results unify outwardly distinct representations across unsupervised learning, network science, and imaging neuroscience.

## 1 Introduction

Many measures of statistical association reduce to variants of the dot product. For example, the covariance is the dot product of mean-centered vectors, the cosine similarity is the dot product of normalized vectors, and the Pearson correlation coefficient is the dot product of mean-centered and normalized vectors.

Here, we consider a parallel reduction of several representation methods to variants of a generalized matrix dot product (or Frobenius inner product). We consider a symmetric non-negative similarity matrix $\mathbf{C} \in \mathbb{R}_{\geq 0}^{n \times n}$, its low-dimensional representation $\mathbf{U} \in \mathbb{R}^{n \times k}$, with $k \ll n$, and a matrix dot product of the form

$$\text{tr}(f(\mathbf{C})^{\top} g(\mathbf{U})) = \sum_{i,j}[f(\mathbf{C})_{ij}][g(\mathbf{U})_{ij}], \tag{1}$$

where $f : \mathbb{R}_{\geq 0}^{n \times n} \to \mathbb{R}^{n \times n}$ is a transform of $\mathbf{C}$, and $g : \mathbb{R}^{n \times k} \to \mathbb{R}^{n \times n}$ is a similarity of $\mathbf{U}$.

We use this formalism to equate several popular but outwardly distinct objectives across unsupervised learning, network science, and imaging neuroscience. The equivalences we describe are algebraically simple, but we have not previously encountered them in the unsupervised-learning or network-science literatures. For convenience, we anchor our discussion around the objective of network modularity.

## 2 Unified modularity and $k$-means objective

The modularity is probably the most popular objective in network science [11]. We begin by equating this method with the $k$-means objective, probably the most popular clustering objective in unsupervised learning [14]. It is difficult to precisely quantify the number of all studies that have sought to optimize these objectives. Nonetheless, citation numbers of canonical references [3, 16, 18, 22] suggest an estimate in the tens of thousands of studies for each objective.

The two methods have strong conceptual similarities insofar as both seek to find groups of similar elements in data or networks. Thus, $k$-means clustering seeks to find data clusters that have relatively

Preprint.

high within-cluster similarity. Similarly, modularity maximization seeks to find network modules that have relatively high within-module connectivity.

Despite these similarities, the standard literature usually views the modularity and the $k$-means objective to be distinct [11], or relates them only in narrow and somewhat abstract regimes [15]. Here, by contrast, we show that these objectives form closely related variants of Equation 1. We do so by showing an equivalence between a normalized modularity and a centered $k$-means objective.

We will sketch out this equivalence by focusing on the modularity. For some network matrix $\mathbf{C}$ and binary module indicator matrix $\mathbf{U}$, we can define this objective, up to a rescaling constant, as

$$\text{(modularity)} \equiv \text{tr}\left[\left(\mathbf{C} - \frac{\mathbf{d}\mathbf{d}^\top}{\mathbf{d}^\top\mathbf{1}}\right)\left(\mathbf{U}\mathbf{U}^\top\right)\right] \tag{2}$$

where $\mathbf{1}$ is the $n$-length vector, and $\mathbf{d} \in \mathbb{R}^{n \times 1}$ is a vector of weighted node degrees, $\mathbf{d} = \mathbf{C}\mathbf{1}$.

We can now describe the basis of the equivalence.

First, we consider the role of the term $\left(\mathbf{d}\mathbf{d}^\top\right)/\left(\mathbf{d}^\top\mathbf{1}\right)$ in Equation 2. This term is often viewed as a null model, or baseline, for assessing the quality of the module partition. This common view is problematic, however, because a null model is usually defined by a null distribution, rather than merely by the expectation of this distribution. This difference is important because the null distribution allows us to estimate statistical uncertainty and significance for each partition, while the expectation alone does not. Correspondingly, treatment of the expectation as a type of null model in network science has led to paradoxical reports of modular structure in seemingly non-modular networks [12, 23].

Here, by contrast, we view the expectation term from an alternative and more defensible perspective. We note, specifically, that subtraction of this term is equivalent to a double centering of $\mathbf{C}$. Formally, we can define a centering operator $\mathbf{H} = \mathbf{I} - (\mathbf{d}\mathbf{1}^\top)/(\mathbf{d}^\top\mathbf{1})$ and use it to rewrite Equation 2 as

$$\text{(modularity)} \equiv \text{tr}\left[\left(\mathbf{H}\mathbf{C}\mathbf{H}^\top\right)\left(\mathbf{U}\mathbf{U}^\top\right)\right]. \tag{3}$$

Next, we define a new variant of the modularity that normalizes the contribution of each module by its size, or the number of its constituent nodes. We term this normalized variant the $k$-modularity. The normalization in $k$-modularity gives more importance to dense modules, rather than merely to large modules, and thus intuitively aligns with the general aim of the modularity. Formally, we can define a normalization operator $\mathbf{N} = \text{diag}(N_1, N_2, \ldots, N_k)$, where $N_h$ denotes the number of nodes in module $h$. We can then use $\mathbf{N}$ to define the $k$-modularity as

$$(k\text{-modularity}) \equiv \text{tr}\left[\left(\mathbf{H}\mathbf{C}\mathbf{H}^\top\right)\left(\mathbf{U}\mathbf{N}^{-1}\mathbf{U}^\top\right)\right]. \tag{4}$$

Finally, let us define a data matrix $\mathbf{X} \in \mathbb{R}^{t \times n}$ with constant column norms, $\|\mathbf{x}_1\| = \|\mathbf{x}_1\| \cdots = \|\mathbf{x}_k\|$, and let us suppose that $\mathbf{C} = \mathbf{X}^\top\mathbf{X}$. Under this definition, we can easily express the standard $k$-means objective computed on $\mathbf{X}$ as

$$(k\text{-means objective}) \equiv \text{tr}\left[\mathbf{C}\left(\mathbf{U}\mathbf{N}^{-1}\mathbf{U}^\top\right)\right]. \tag{5}$$

Together, these results establish that the modularity and the $k$-means objective form special cases of a single general objective. The main difference between these objectives lies in data transformation or feature constraints, rather than in more fundamental distinctions. The modularity centers the data, while $k$-means normalizes the sum of within-module weights. The $k$-modularity unifies $k$-means and modularity by combining both properties in one objective. By analogy with our discussion of the vector dot product in the Introduction, these relationships roughly align the modularity with the covariance, the $k$-means objective with the cosine similarity, and the $k$-modularity with the Pearson correlation coefficient.

## 3  Unified modularity and spectral manifold learning

We begin by noting the following three results. First, a well-known result in linear algebra [10] states that the $k$ leading eigenvectors $\mathbf{u}_1, \mathbf{u}_2, \ldots, \mathbf{u}_k$ of some symmetric matrix $f(\mathbf{C})$ maximize the sum

$$\sum_{h=1}^{k} \frac{\mathbf{u}_h^\top f(\mathbf{C})\mathbf{u}_h}{\mathbf{u}_h^\top \mathbf{u}_h}, \tag{6}$$

relative to any other set of $k$ orthogonal vectors. Each term in this sum is known as a so-called Rayleigh quotient. Here, we consider a variant of this sum with binary constraints on the values of $\mathbf{u}_h$. Under these constraints, the number of nonzero elements in $\mathbf{u}_h$ is given simply by $N_h = \mathbf{u}_h^\top \mathbf{u}_h$. This immediately allows us to reduce the objective in Equation 6 to the $k$-means objective in Equation 5.

Second, we note that the degree of (native or transformed) correlation networks $\mathbf{d}$ usually aligns with the leading eigenvector of these networks [13, 17]. After denoting this leading eigenvector by $\mathbf{v}_1$ and its eigenvalue by $\lambda_1$, we can establish the following approximate equivalence:

$$\mathbf{C} - \frac{\mathbf{d}\mathbf{d}^\top}{\mathbf{d}^\top \mathbf{1}} \approx \mathbf{C} - \lambda_1 \mathbf{v}_1 \mathbf{v}_1^\top. \tag{7}$$

Third, the first two results combined with the definition of $k$-modularity in Equation 4, suggest that the detection of $k$-modularity modules is approximately equivalent to the detection of second to $(k+1)^{\text{th}}$ leading eigenvectors with binary constraints or, alternatively, to the detection of $k$ binarized leading eigenvectors after rank-1 subtraction.

We now note that several prominent spectral manifold learning methods, including Laplacian eigenmaps [2], diffusion maps [5], and local linear embedding [25] reduce to the detection of the $k$ leading eigenvectors of some (method-specific) $f(\mathbf{C})$ after rank-1 subtraction. These commonalities thus imply an approximate equivalence between the $k$-modularity of $f(\mathbf{C})$ and binary variants of these spectral learning methods. Figure 1 shows an example illustration of this approximate equivalence on functional-MRI networks.

## 4   Unified modularity and UMAP

We finally define m-umap, a generalized modularity objective, and relate it to UMAP [20], a prominent manifold-learning method especially popular in the analysis and visualization of single-cell, population-genetic, and other biological data [1, 8]. We specifically derive m-umap as a first-order approximation of UMAP, describe that the binary variant of m-umap reduces to the modularity, and show that m-umap embeds example brain-imaging data better than UMAP.

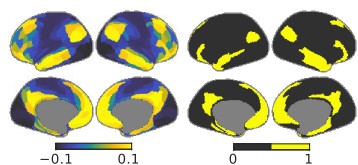

Figure 1: Diffusion-map components (functional gradients in imaging neuroscience [19], left) and $k$-modularity modules of correlation networks (right).

The performance of UMAP was originally attributed to its deep theoretical underpinnings [20]. More recently, however, a sequence of intriguing empirical and analytical work [4, 7, 6] has shown that the standard implementation of UMAP deviates from its motivating theory. This work has shown that the true UMAP objective — the objective actually optimized by the standard implementation of this method [21] — essentially seeks to find low-dimensional representations that accurately approximate symmetric $\kappa$-nearest-neighbor network matrices. It does so by nonlinearly aligning these representations to the nearest-neighbor networks, using a so-called Cauchy similarity, a measure of similarity that pushes nodes away even at long ranges and thus prevents the formation of amorphous clouds or network hairballs.

In this section, we denote the binary symmetric $\kappa$-nearest-neighbor network matrix by $\mathfrak{C} \in \{0,1\}^{n \times n}$, its degree vector by $\mathfrak{d}$, and the corresponding centering operator by $\mathfrak{H}$. We now consider a parametric implementation of UMAP [26]. Damrich and Hamprecht [7] showed that, up to a rescaling constant, the true parametric UMAP objective can essentially be written as

$$(\text{UMAP objective}) = -\sum_{i,j} \left( \mathfrak{c}_{ij} \log(\phi_{ij}) + \gamma \frac{\mathfrak{d}_i \mathfrak{d}_j}{\mathfrak{d}^\top \mathbf{1}} \log\left(1 - \phi_{ij}\right) \right),$$

where $\sum(\mathfrak{c})$ denotes the sum of all elements of $\mathfrak{C}$, the Cauchy similarity

$$\phi_{ij} = \left(1 + \alpha \left\| \mathbf{u}_{i:} - \mathbf{u}_{j:} \right\|^{2\beta}\right)^{-1}$$

is a function of the Euclidean distance between embedding vectors $\mathbf{u}_{i:}$ and $\mathbf{u}_{j:}$, and $\alpha$, $\beta$, and $\gamma$ are parameters (note that we use $\mathbf{u}_{i:}$ to denote row vectors).

We define m-umap as the first-order Taylor expansion of the UMAP objective around $1/2$. After doing the algebra and dropping all constant terms, we can write

$$\text{(m-umap)} = -\sum_{i,j} \left( \mathfrak{c}_{ij} - \gamma \frac{\mathfrak{d}_i \mathfrak{d}_j}{\mathfrak{d}^\top \mathbf{1}} \right) \phi_{ij} = -\operatorname{tr}\left[ \left( \mathfrak{H} \mathfrak{C} \mathfrak{H}^\top \right) \Phi \left( \mathbf{U} \right) \right], \tag{8}$$

where we use $\Phi(\mathbf{U})$ to denote the pairwise Cauchy-similarity matrix of the rows of $\mathbf{U}$.

Clearly, therefore, m-umap is just the modularity of symmetric $\kappa$-nearest-neighbor matrices with Cauchy similarity. For completeness, we also note that $\gamma$ in Equation 8 denotes the so-called modularity resolution parameter [24], and corresponds exactly to the so-called negative-sampling rate of UMAP [7].

Both m-umap and UMAP thus seek to approximate symmetric $\kappa$-nearest-neighbor representations and coincide in their theoretical optimum. We can only get to this optimum, however, if we place disconnected nodes infinitely far apart. In more realistic settings, m-umap is simpler than UMAP and places more emphasis on fidelity over aesthetics. For example, m-umap always pulls connected nodes together and pushes disconnected nodes apart. By contrast, UMAP pulls connected nodes together if they are far apart but also pushes them apart if they are too close together (and thus prevents their collapse into single points).

The simplicity of m-umap can lead to runaway solutions, such as the infinite repulsion of disconnected nodes. Here, we check this outcome by embedding m-umap solutions on ($k$-dimensional) spheres. The spherically embedded m-umap forces each row vector $\mathbf{u}_{i:}$ to have unit-norm and implies that $\|\mathbf{u}_{i:} - \mathbf{u}_{j:}\|^2 = 2\left(1 - \mathbf{u}_{i:}^\top \mathbf{u}_{j:}\right)$. Under the additional assumptions of binary $\mathbf{u}_{i:}$ we have $\phi_{ij} = 1$ when $\mathbf{u}_{i:} = \mathbf{u}_{j:}$ and a constant $0 < \phi_{ij} < 1$ otherwise. It is easy to show that this additional constraint reduces the m-umap objective to the standard modularity of $\mathfrak{C}$.

To directly compare m-umap and UMAP, we optimized these objectives to derive cartographic representations of a 59,412 voxel-level functional-MRI correlation network, constructed from resting-state functional MRI scans of 100 people in the Human Connectome Project [9]. Despite the general popularity of UMAP, we could not find a published example of such a representation in the imaging-neuroscience literature. This hints perhaps at a basic challenge of using UMAP on the blurred and sluggish functional MRI data. Correspondingly, in our hands, UMAP consistently produced cloud representations with little discernible intrinsic morphology (Figure 2, left row).

By contrast, m-umap produced much better embeddings of these data (Figure 2, right row). It clearly separated neurobiological modules that were derived with modularity maximization of symmetric $\kappa$-nearest-neighbor networks (binary m-umap). Quantitatively, m-umap produced embeddings with $\approx 8\%$ higher fidelity than UMAP. Together, these results suggest that m-umap's mix of discrete module detection and continuous embedding allows it to successfully capture the similar mix of discrete and continuous aspects of cortical organization.

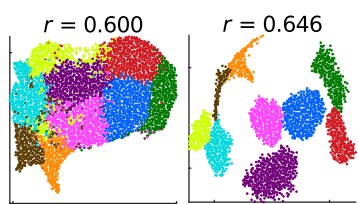

Figure 2: UMAP (left) and m-umap (right) embeddings of highly resolved correlation networks. Colors denote module identities estimated with binary m-umap (i.e., modularity maximization). $r$ values represent Pearson correlation coefficients between distances of nodes to module centroids, in native and embedding space.

## 5    Conclusion

We reduced diverse and outwardly distinct representation objectives to variants of the modularity and, more generally, the Frobenius inner product (Equation 1). From this perspective, we found that many objectives blend into each other through changes in data transforms and feature definitions. Our results suggest that a principled motivation for choosing these transforms and features in individual studies is likely to be more important than the choice of outwardly distinct but inwardly equivalent methods. More generally, we hope that this unifying perspective helps to consolidate methodology across unsupervised learning, network science, and imaging neuroscience.

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
