# OpenReview forum: "Equivalences between network modularity and diverse low-dimensional representations"
_NeurIPS.cc/2025/Workshop/UniReps — UniReps2025_

### Official Review · Reviewer_ZMK2 · 2025-09-13
**Connections between Modularity, Classical Algorithms and UMAP**

**Confidence:** 4

**Review:**

**Summary**

The work investigates the equivalence between modularity (clustering algorithm in network science) and classical algorithms such as k-mean, spectral manifold learning, and UMAP. It then proposes a generalized modularity objective  *m-umap* . Prelimary result on neuroscience data shows m-umap leads embedding with higher fidelity.


**Comments:**
- I find the connections between discussed methods interesting.
	- Formal proof steps might further aid accecibility
- L80: what does binarization mean?
- L130-144: although the result in Fig 1 is promising, the superiority of m-umap could be more if the work could demonstrates that on published UMAP results.
- L151: it might be better if there is a sentence on how the fidelity is computed.

**Score:**

4

**Topic Fit:**

1

---

### Official Review · Reviewer_gkoM · 2025-09-16
**Strong theoretical work!**

**Confidence:** 4

**Review:**

Similarity metrics are myriad and have found uses in disparate fields. Here the authors show that these myriad similarity metrics are more similar to each other than previously thought. The authors show that a large class of commonly used metrics can be thought of a generalized dot products. This formulation enabled the authors to draw a strong connection between to similarity metrics as seemingly different as network modularity and a normalized version of k-means clustering. Beyond elucidating this connection, the authors begin to demonstrate how this newly found insight can be used to develop new similarity metrics better tailored to the needs of their specific domains. For instance, developing a modified version of the u-map algorithm that better respects global geometry.

The hallmark of a clearly written paper is that the ideas introduced seem obvious after the fact. After reading this it now seems obvious that k-modularity and k-means should be so deeply connected. I think this finding enables the theoretical study of a large class of these generalized dot products. I think this work can do a little more regarding the optimal form or in this case, normalization, should be applied to different similarity metrics. For example, does centering tend to have a systematic effect on a similarity metric’s ability to capture global aspects of geometry? M-UMAP is effectively a first order taylor approximation of the commonly implemented UMAP; a common challenge with manifold learning methods is the difficulty of interpreting their hyperparameters. I think there is a good chance that this line of work will do a lot to clarify what these hyperparameters are doing and to help practitioners choose a suitable set of hyperparameters for their task.

**Score:**

5

**Topic Fit:**

3

---

### Official Review · Reviewer_xPWd · 2025-09-16
**Review of Equivalences between network modularity and diverse low-dimensional representations**

**Confidence:** 3

**Review:**

Clustering data based on similarity of samples is an important problem in many domains, including representation analysis. In this work, the authors demonstrate that many different similarity measures, including k-means clustering and UMAP, fall under a single objective of network modularity. Specifically, they propose an objective based on the Frobenius inner product between a network matrix C and module indicator U. They begin by showing that modularity in network science and k-means is a special cases of this generalized objective. Next, they show that standard implementations of UMAP also follow under this umbrella. They propose m-umap as Taylor approximation to UMAP, and show that it optimizes a similar objective. They suggest that their objective is simpler and has greater fidelity than UMAP, and they verify this with preliminary work on functional MRI data.

**Strengths**
- Well, written and easy to follow
- Unifying various clustering objectives is an important contribution to many domains

**Weaknesses**
- The authors suggest that their results place a greater emphasis on data transformations and features over clustering objectives, but a reader will likely benefit from concrete examples of this claim
- If this work goes beyond an extended abstract, more evaluation is needed, and I would start with pedagogical examples w.r.t. my point above

**Score:**

4

**Topic Fit:**

3

---

### Official Review · Reviewer_2F9d · 2025-09-16
**Review of "Equivalences between network modularity and diverse low-dimensional representations"**

**Confidence:** 3

**Review:**

The authors present a connection between modularity and three other sets of objectives. While the writing is clear and easy to follow, it does get a little hand-wavy at some points.

Strengths:

1) The paper tries to unify concepts across different domains.

2) The writing is clear.

Weaknesses/suggestions:

1) Section 2 is underwhelming. K-means is a popular network clustering objective too, and the correspondence seems unsurpising. I wouldn’t call these ‘outwardly distinct’.

2) There’s only one figure in the paper, yet the authors refer to Figure 4 top row and Figure 4 bottom row (that is, the figure is mislabeled).

3) Only UMAP and m-UMAP are compared while all the metrics the authors claim are equivalent (at least approximately) should be compared.

4) Section 3 is also underwhelming and a little hand-wavy. It would have been better to include some results comparing the techniques here.

5) I am not sure about the fit between the paper and the workshop theme. While I am recommending acceptance, I believe that the authors will get better feedback at a network science conference.

**Score:**

3

**Topic Fit:**

2